# Non-Polio Enterovirus C Replicate in Both Airway and Intestine Organotypic Cultures

**DOI:** 10.3390/v15091823

**Published:** 2023-08-27

**Authors:** Giulia Moreni, Hetty van Eijk, Gerrit Koen, Nina Johannesson, Carlemi Calitz, Kimberley Benschop, Jeroen Cremer, Dasja Pajkrt, Adithya Sridhar, Katja Wolthers

**Affiliations:** 1OrganoVIR Labs, Department of Medical Microbiology, Amsterdam UMC, Location AMC, Amsterdam Institute for Infection and Immunity, University of Amsterdam, 1105 AZ Amsterdam, The Netherlands; h.w.vaneijk@amsterdamumc.nl (H.v.E.); g.koen@amsterdamumc.nl (G.K.); n.a.johannesson@amsterdamumc.nl (N.J.); c.calitz@amsterdamumc.nl (C.C.); a.sridhar@amsterdamumc.nl (A.S.); k.c.wolthers@amsterdamumc.nl (K.W.); 2OrganoVIR Labs, Department of Pediatric Infectious Diseases, Emma Children’s Hospital, Amsterdam UMC, Location AMC, University of Amsterdam, 1105 AZ Amsterdam, The Netherlands; d.pajkrt@amsterdamumc.nl; 3National Institute for Public Health and Environment, RIVM, 3721 MA Bilthoven, The Netherlands; kim.benschop@rivm.nl (K.B.); jeroen.cremer@rivm.nl (J.C.)

**Keywords:** non-polio enterovirus, enterovirus C species, tropism, CVA-13, CVA-20, EV-C99, organotypic cultures, organoids, human airway epithelium, human intestinal epithelium

## Abstract

Non-polio enteroviruses (EV) belonging to species C, which are highly prevalent in Africa, mainly among children, are poorly characterized, and their pathogenesis is mostly unknown as they are difficult to culture. In this study, human airway and intestinal organotypic models were used to investigate tissue and cellular tropism of three EV-C genotypes, EV-C99, CVA-13, and CVA-20. Clinical isolates were obtained within the two passages of culture on Caco2 cells, and all three viruses were replicated in both the human airway and intestinal organotypic cultures. We did not observe differences in viral replication between fetal and adult tissue that could potentially explain the preferential infection of infants by EV-C genotypes. Infection of the airway and the intestinal cultures indicates that they both can serve as entry sites for non-polio EV-C. Ciliated airway cells and enterocytes are the target of infection for all three viruses, as well as enteroendocrine cells for EV-C99.

## 1. Introduction

Enteroviruses (EV) are small non-enveloped viruses with a 30 nm diameter icosahedral capsid containing a positive sense single-stranded RNA (ssRNA) genome. Together with Rhinoviruses (RV), they belong to the *Enterovirus* genus within the *Picornaviridae* family. Human enteroviruses comprise of more than 100 different types divided into four (EV A-D) of the 15 (EV A-L) enterovirus species identified [1]. EVs are highly prevalent worldwide with millions of cases every year, entailing a heavy economic burden globally [2,3]. They can infect different organs, resulting in a wide variety of clinical presentations in both children and adults. Symptoms can range from a common cold, rash, and mild respiratory and gastrointestinal symptoms to more severe conditions like meningitis, encephalitis, acute paralysis, neonatal sepsis, myocarditis, and hepatitis [4,5,6,7,8]. Their widespread presence and their extensive disease range are a consequence of the significant genomic variability exhibited by EVs, as they are subject to a high number of mutations and recombination [9]. Inter-species recombination within the EV-C species has shown to be particularly crucial in the emergence of novel neuropathogenic vaccine-derived polioviruses (VDPVs) [10,11,12,13]. The results of these two evolutionary processes can lead to substantial alterations in viral pathogenesis, host tropism, and infection outbreaks [14,15]. Therefore, studying non-polio EV-C is critical when considering their potential threat to poliovirus eradication [12,16].

Little is known about non-polio EV-C pathogenesis and disease presentation due to their poor cultivability in standard immortalized cell lines, low-case reports, and limited surveillance. Although in most cases, infection with non-polio EV-C does not cause any symptoms, their disease burden should not be underestimated since these viruses have also been repeatedly associated with cases of acute flaccid paralysis (AFP) [17]. These hard-to-grow viruses are predominantly present in Africa, and they spread mainly among children [18,19]. Based on the studies on poliovirus [20,21,22] and other EV [23,24], the primary replication sites of EV-C are believed to be the gastrointestinal and respiratory epithelium. CVA-1, CVA-13, CVA-15, and CVA-17 to CVA-22, EV-C96, and EV-C99 are considered gastrointestinal EV-C as they are mainly isolated from the stool samples in cases of mild to severe gastrointestinal symptoms [25,26]. The most recently discovered EV-C, namely EV-C104, 105, 109, 116, 117, and 118, are almost exclusively isolated from the respiratory samples of patients with respiratory symptoms, and therefore, commonly identified as respiratory EV-C [25,26].

The most common EV-C detected in Africa is EV-C99, followed by CVA-13 and CVA-20 [19,27,28,29]. These three EV-C have also been identified in the isolates from children in China [30], as well as in the samples from sewage or from immunocompromised patients in Europe [31,32]. To elucidate the tropism of non-polio EV-C genotypes in these replication sites at both the tissue and cellular level, we here aim to study EV-C99, CVA-13, and CVA-20 using human organotypic models. These human organotypic models have been implemented to be fully exploited in viral studies [33]. In this work, we used a Transwell^®^-based model of the human airway and human intestine that enables access to both the apical and basolateral sides of the cultures. These cultures contain differentiated cell types of the respective organ, which include ciliated cells, goblet cells, and basal cells for the human airway epithelium (HAE) [34,35], and enterocytes, enteroendocrine cells, goblet cells, Paneth cells, and stem cells for the human intestinal epithelium (HIE) [36,37,38,39]. Moreover, we compared infection in fetal-derived organotypic models versus adult-derived organotypic models in order to investigate whether the differences in efficacy or the tropism of infection plays a role in the higher prevalence of EV-C in children as compared to adults.

## 2. Materials and Methods

### 2.1. Cell Lines and Viruses

Caco2 cells (human colon adenocarcinoma; kindly provided by Leiden University Medical Center, Leiden, the Netherlands) were maintained in Eagle’s minimum essential medium (EMEM, Lonza, Basel, Switzerland) supplemented with 0.1% (*v*/*v*) L-glutamine (Lonza, Basel, Switzerland), 1% (*v*/*v*) non-essential amino acids (100×, ScienceCell Research Laboratories, CA, USA), 100 U/mL of penicillin and 100 U/mL of streptomycin (Pen-Strep, Lonza, Basel, Switzerland), and 20% (*v*/*v*) heat-inactivated fetal bovine serum (FBS, Sigma-Aldrich, St. Louis, MI, USA). Cells were cultured at 37 °C, 5% CO_2_, and passaged every week. CVA-13, CVA-20, and EV-C99 were isolated on Caco2 cells directly from the stool samples originating from 2001; kindly provided by prof. M. Boele van Hensbroek from the Amsterdam UMC. Samples had been anonymized, and culturing was performed after the determination of the virus genotype via Sanger sequencing of the VP1 region. Viral stocks were obtained based on the presence of a cytopathic effect (CPE). Two passages were performed: the first passage was required to amplify the virus in the culture, and the second was to create a large, concentrated stock of the virus. In both passages, a complete CPE was observed. In the viral stocks of CVA-13, CVA-20, and EV-C99 cultured on Caco2, the cell lines were used further in this study for infection in the human airway epithelial (HAE) and intestinal epithelial (HIE) cultures. The inoculation of HIE directly with the clinical specimens was unsuccessful, and due to the limited amount of original material, further attempts to culture them directly were not undertaken. On the other hand, CVA-13 and EV-C99 from the original fecal material replicated when directly inoculated on HAE. However, a small volume was retrieved from these cultures, making it difficult to use it for further experiments.

### 2.2. Next-Generation Sequencing

Original stool samples and their respective clinical isolates of CVA-13, CVA-20, and EV-C99 were sequenced using a previously published protocol [40]. Briefly, MagNA Pure 96 (Roche Diagnostics, Basel, Switzerland) was used for RNA extraction; Nextera XT DNA Library Preparation Kit (Illumina, San Diego, CA, USA) for tagmentation and library preparation; and Nextseq (Illumina, San Diego, USA) for the sequencing runs. Samples cultured directly on HAE were sequenced using MagNA Pure 96 (Roche Diagnostics, Basel, Switzerland) for DNA/RNA extraction, DNase treatment, and RNA concentration with RNA Clean & Concentrator™-5 (Zymo research, Irvine, USA). The Illumina DNA Prep kit (Illumina, San Diego, USA) was used for tagmentation and library preparation and Nextseq (Illumina, San Diego, USA) for the sequencing runs. The full-length sequences were obtained using the Jovian platform (https://jovian.rivm-bioinformatics.com, web based) and Genome Detective v1.139 (https://www.genomedetective.com/app/typingtool/virus/, web-based). The obtained full-length sequences were aligned using CodonCode Aligner software version 9.0.2 (CodonCode Corporation, Centerville, OH, USA). Based on the complete sequence of the VP1 region for all three viruses, the typing of the original fecal material and the clinical isolates was confirmed.

### 2.3. Primary Fetal Airway Epithelial Cell Isolation

Primary fetal airway epithelial cells were isolated from the trachea and bronchi of the respiratory tract from anonymous fetal donors (gestational age 17–20 weeks, gender unknown). The isolation was performed based on a previous publication on adult tissue [41] and adapted for fetal tissue. After removing the excess connective tissue, tissue was cut into small pieces of about one cm and washed three times with a wash medium prepared using Joklik’s minimum essential medium (JMEM) supplemented with 1% (*v*/*v*) L-Glutamine, 50 µg/mL of gentamycin, 0.25 µg/mL of amphotericin B, 100 U/mL of penicillin, 100 U/mL of streptomycin, and 100 U/mL of nystatin. All reagents were from Sigma-Aldrich, St. Louis, MO, USA. Clean tissue was transferred to a 15 mL tube and digested in a 1:9 dilution of a protease/Dnase solution in the wash medium overnight at 4 °C on a tube rotator. The Protease/Dnase solution was prepared using a cold phosphate-buffered saline (PBS, Lonza, Basel, Switzerland) with the addition of 1 mg/mL of protease XIV (Sigma-Aldrich, St. Louis, MO, USA) and 10 µg/mL of Dnase (Sigma-Aldrich, St. Louis, MO, USA). Tissue dissociation was ended by adding 10% FBS and undigested tissue was discarded. After centrifugation at 500× *g* for 5 min at 4 °C, the pellet obtained was washed once, and then cells were seeded on a collagen-coated T75 flask (rat tail collagen type I, Ibidi, Gräfelfing, Germany) with PneumaCult™-Ex Plus Medium (STEMCELL™ Technologies, Cambridge, UK) supplemented with 50 µg/mL of gentamycin (Sigma-Aldrich, St. Louis, USA), 0.25 µg/mL of amphotericin B (Sigma-Aldrich, St. Louis, USA), 100 U/mL of penicillin, 100 U/mL of streptomycin (Sigma-Aldrich, St. Louis, USA), 100 U/mL of nystatin (Sigma-Aldrich, St. Louis, USA), and 10 µM of Rho-kinase (Rock) inhibitor Y-27632 (Sigma-Aldrich, St. Louis, USA). Medium change was performed the day after isolation and then every two days. After three days, cells were maintained in PneumaCult™-Ex Plus Medium supplemented only with 1% (*v*/*v*) Pen-Strep.

### 2.4. Fetal and Adult Human Airway Epithelial (HAE) Cultures

Once the primary fetal airway cells reached 60–80% confluence, they were detached and dissociated into single cells using Trypsin/EDTA Solution (Lonza, Basel, Switzerland) and seeded on pre-coated Transwell^®^ inserts consisting of a porous polyester (PET) membrane with a pore diameter size of 0.4 µm (HTS Transwell^®^ 24-well plate, Corning, New York, USA). Transwell coating was performed by adding 100 µL of rat tail collagen type I diluted in 0.1% (*v*/*v*) acetic acid (VWR Chemicals, Radnor, PA, USA) to a final concentration of 20 µg/mL, incubating for an hour at room temperature (RT), and subsequently washing three times with PBS before cell seeding. Primary fetal airway cells were then seeded on the apical compartment at a cell density of 8 × 10^4^ cells/insert in a 200 µL volume, and 500 µL of the medium was added to the basolateral compartment. Cells were supplemented with 10 µM of the Rock inhibitor for the first three days and maintained in 1% Pen-Strep PneumaCult™-Ex Plus Medium until they formed a complete cell monolayer (2–5 days). After the expansion phase was completed, the HAE cultures were differentiated by exposing the apical side to air through a transition from a liquid–liquid interphase (LLI) to an air-liquid interphase (ALI) and replacing the basolateral medium with 1% (*v*/*v*) Pen-Strep PneumaCult™-ALI Medium (STEMCELL™ Technologies, Cambridge, UK). Medium change was performed every 2–3 days for 28 days. Monolayer integrity was assessed by measuring the trans-epithelial electrical resistance (TEER) using a voltohmmeter device (EVOM2™, World Precision Instruments, Sarasota, USA). TEER measurements were conducted in triplicate per each insert by placing the STX2 chopstick electrodes in the three openings available for each Transwell™. A coated insert without cells was used to determine the baseline resistance value (Ω). TEER values were then calculated by subtracting the baseline resistance and multiplying it by the surface area (cm^2^) of the membrane. Only inserts with TEER values above 200 Ω × cm^2^ were used for infection. The adult HAE cultures, MucilAir™, from a pool of 14 donors, were obtained from Epithelix Sàrl (Geneva, Switzerland)

### 2.5. Fetal Small Intestinal Cultures

For the generation of fetal small intestinal organoids, crypts were isolated from the small intestinal tissue of the anonymous fetal donors (gestational age 14–17 weeks, gender unknown). as described previously [39]. Isolated crypts were suspended in Matrigel^®^ (Corning, New York, NY, USA), dispensed in three 10 μL droplets per well in a 24-well tissue culture plate, and covered with a 500 μL medium. The enteroid cultures were maintained at 37 °C, 5% CO_2,_ in Human IntestiCult™ Organoid Growth Medium (STEMCELL™ Technologies, Cambridge, UK) supplemented with 100 U/mL of Pen-Strep. The medium was replenished every other day, and the organoids were passaged every 3–5 days as described previously [42]. Adult primary small intestine epithelial cells (Lonza Bioscience, Walkersville, MA, USA) from three donors (751357, 751359, and 751360) were used to obtain enteroids following the same protocol used for fetal cells.

### 2.6. Fetal and Adult Human Intestinal Epithelial (HIE) Cultures

Transwell™ (HTS Transwell 24-well plate, Corning, New York, NY, USA) were coated with rat tail collagen type I in the same way described earlier for HAE. Human fetal and adult enteroids were collected and a single cell suspension was obtained via treatment with TrypLE™ (Gibco, Thermo Fisher Scientific, Waltham, USA) for 10 min at 37 °C. Cells were diluted to 10^6^ cells/mL in Human IntestiCult™ Organoid Growth Medium (OGMh) (STEMCELL™ Technologies, Cambridge, UK) supplemented with 10µM of the Rock inhibitor and 100 μL of cell suspension was seeded per each insert. Organoids were expanded for 7 days in OGMh to form monolayers. On day 7, differentiation was initiated by using IntestiCult™ OGMh Differentiation Medium obtained by adding IntestiCult™ OGMh Basal Medium and Advanced Dulbecco’s Modified Eagle Medium/Nutrient Mixture F-12 (DMEM/F12, Gibco, Thermo Fisher Scientific, Waltham, USA) in a 1:1 ratio supplemented with 1% (*v*/*v*) Glutamax (Thermo Fisher Scientific, Waltham, USA), 1% (*v*/*v*) Pen-Strep, and 1% (*v*/*v*) Hepes (Sigma-Aldrich, St. Louis, USA). Monolayers were differentiated for 7 days until day 14 and the medium was replenished every other day. To assess monolayer integrity, TEER was measured on days 7, 10, and 14, and only inserts with TEER values at day 14 above 200 Ω × cm^2^ were used for infection.

### 2.7. HAE and HIE Infection

MucilAir™ adult HAE cultures from a pool of 14 donors were infected in duplicates while all the other cultures from the three donors each were infected in triplicates. The multiplicity of infection (MOI) was calculated based on the average number of cells/HAE insert (5 × 10^5^) or cells/HIE insert (2 × 10^5^) and used for the infection of both the fetal and adult HAE and HIE cultures. After washing the apical compartment with 100 µL of the medium to remove the excess mucus, the HAE and HIE inserts were infected by adding 50 µL of CVA-13, CVA-20, or EV-C99 (MOI = 0.02) to the apical side or 50 µL to the basolateral side together with 450 µL of the medium. Infected HAE and HIE were incubated at 37 °C with 5% CO_2_ for 2 h. After incubation, monolayers were washed three times with Advanced DMEM/F12 on both sides. Fresh medium was added to the apical (100 µL) and basolateral compartment (500 µL), and the cultures were incubated at 37 °C with 5% CO_2_ for 10 min. After ten minutes, 100 µL of the medium from each compartment were collected for the first time point (0 h) and immediately replenished with fresh medium, allowing for the following time point collection except for the HAE cultures that were left at the ALI interface, and upon sampling, 100 µL of the medium was added to the apical compartment and collected after 10 min of incubation at 37 °C. Medium collection was carried out at 24, 48, 72, 96, and 120 h post-infection (hpi). Non-infected inserts in triplicates were included in the experiment as negative controls.

### 2.8. Quantitative RT-qPCR to Determine Viral Copy Numbers

Total viral RNA was purified using the PureLink™ RNA Mini Kit (Thermo Fisher Scientific, Waltham, USA) according to the supplier’s protocol. A lysis of 25 µL of the medium sample was performed in 300 µL of the lysis buffer. Reverse transcription to cDNA was carried out on 40 µL of the eluted RNA. A quantitative real-time polymerase chain reaction (RT-qPCR) was performed on 5 µL of the cDNA sample using 1× conc. LightCycler^®^ 480 SYBR Green I Master (Roche, Basel, Switzerland), 0.25 µM of pan-Enterovirus forward primer (5′-*GG*CCCTGAATGCGGCTAAT-3′, 452–468), and 0.25 µM of pan-Enterovirus reverse primer (5′-*GGG*ATTGTCACCATAAGCAGCC-3′, 579–597) (Biolegio, Nijmegen, the Netherlands) designed against the 5′ untranslated (5′UTR) region of the Enterovirus genome according to a previous publication [43]. qPCR was performed in a CFX96 Connect Real-Time PCR Detection System (Bio-Rad, Hercules, USA) and viral copies were calculated from the Cq values using a standard curve with known concentrations of the viral genome.

### 2.9. Infectious Viral Particle Estimation via TCID50

The 50% Tissue Culture Infectious Dose (TCID50) of the samples from the apically infected cultures collected at 0 h, 24 h, and 120 h was determined using the Reed and Muench method [44]. Since replication was mainly detected in the apical compartment, infectious viral particles only from the apically collected samples were measured. Titration was performed in the same cell line (Caco2 cells) from which the viruses were isolated and the titers were visualized by normalizing to the 0 h time point.

### 2.10. Immunofluorescence Staining

The HIE and HAE cultures used for immunostaining were apically infected with 100 µL of CVA-13 (TCID50/mL = 7.8), CVA-20 (TCID50/mL = 9.1), or EV-C99 (TCID50/mL = 5.5). Inserts were fixed 8 h post-infection with 4% (*v*/*v*) formaldehyde (Sigma-Aldrich, St. Louis, MO, USA) for 15 min at RT and stored in PBS at 4 °C. Transwells were then permeabilized with 100% ice-cold methanol (VWR Chemicals, Pennsylvania, USA) for 5 min at RT, followed by three washes with PBS. After permeabilization, the membrane was removed from the insert using a scalpel and placed on a glass slide with 200 µL of SEA BLOCK Blocking Buffer (Thermo Fisher Scientific, Waltham, MA, USA). Following overnight blocking at 4 °C, virus staining was performed by adding two drops of the pan-Enterovirus capsid protein VP1 mouse antibody (D^3^ Enterovirus Mab Reagent, 01-050000, Quidel, San Diego, CA, USA). After a 1 h incubation at 37 °C, the VP1 antibody was removed, and the membranes were incubated at 4 °C overnight with 100 µL of the primary antibodies diluted in SEA BLOCK Blocking Buffer (see Appendix A). Following incubation with the primary antibodies, the cultures were washed three times with 1% (*v*/*v*) Tween^®^ 20 (Sigma-Aldrich, St. Louis, MO, USA) in Tris Buffered Saline (TBS) (Merck Millipore, Burlington, MA, USA). Membranes were then incubated with 100 µL of the secondary antibodies (see Appendix A) together with 10 µg/mL of Hoechst 33342 (Thermo Fisher Scientific, Waltham, MA, USA) for nuclei staining for 2 h at RT in the dark. Finally, membranes were mounted using ProLong™ Glass Antifade Mountant (Thermo Fisher Scientific, Waltham, USA), and the slides were imaged with a Leica TCS SP8-X microscope using an HC Plan Apochromat 63× oil objective. After a 0.4 µm thick Z-stacks acquisition, the images were processed with Leica LAS AF Software 4.0.11706 (Leica Microsystems, Wetzlar, Germany), and 3D reconstructions were performed using the LAS-X 3D Software 4.0.11706. At the same time, non-infected membranes were stained as negative controls. Additionally, to ensure the specificity of antibody binding, incubation with secondary antibodies only was also performed.

### 2.11. Statistical Analysis

Experiments were performed in duplicates from a pool of 14 donors in the adult HAE model and in triplicates from three different donors in the other organotypic cultures. The data are presented as mean ± SEM. Statistical significance compared to the 0 h time point was analyzed using two-way ANOVA with Dunnett’s multiple comparison. GraphPad Prism 8 software (GraphPad Software Inc., San Diego, CA, USA) was used for the analysis. *p*-value < 0.05 was considered statistically significant.

## 3. Results

### 3.1. Generating Stocks of Clinical Isolates of CVA-13, CVA-20, and EV-C99

In order to study circulating viruses with clinical relevance, as opposed to lab-adapted viruses, we aimed to work with virus stocks as close as possible to the original specimen. Thus, three fecal samples, previously identified via VP1 sequencing, containing CVA-13, CVA-20, and EV-C99, were taken for virus isolation. We established viral stocks on Caco2 cells and evaluated whether these stocks accumulated mutations over the two passages. Full-genome sequencing of passage 2’s stock was compared to the sequence obtained from the original material. We found that the CVA-13 culture in Caco2 cells resulted in three non-synonymous point mutations. One was observed in the VP3 and two in the non-structural region, but none in the VP1 region (Figure 1). We identified two non-synonymous point mutations in the genome of CVA-20, of which one occurs in the VP1 region. Finally, we showed that, after two passages on Caco2 cells, EV-C99 acquired five non-synonymous mutation: one in the non-coding 5′UTR region, one in the VP3 region, one in the VP1 region, and two in the non-structural region (Figure 1). Although CVA-13 and EV-C99 replicated in the HAE cultures without acquiring any mutations and EV-C99 replicated in the HAE cultures, acquiring four mutations but none in the VP1 region, the volume of the stock virus obtained from these cultures was low. Thus, despite accumulating mutations in the Caco2 cells, these viral stocks were the closest to the original sequence and were considered as clinical isolates to be used for subsequent experiments.

### 3.2. CVA-13, CVA-20, and EV-C99 Replicate in the Human Airway Epithelial (HAE) Model

To study tropism in the primary replication sites, we first infected the human airway epithelial (HAE) cultures with the CVA-13, CVA-20, and EV-C99 clinical isolates. To investigate whether the higher prevalence of non-polio EV-C in infants compared to adults was due to a more pronounced susceptibility of the neonatal airway, we inoculated both fetal, as a neonatal surrogate, and adult-derived HAE cultures (Figure 2). In the adult-derived HAE, when the infection was performed apically, a 100–1000-fold increase in RNA copies was detected in the apical compartment for CVA-13 and CVA-20 with a peak at 1–2 days post-infection (Figure 3a). In the basolateral compartment, CVA-13 and CVA-20 replication kinetics were slower than the apical compartment, but similar RNA levels were reached after two days of infection (Figure 3b). On the other hand, the EV-C99 release was mainly observed basolaterally with more than a 1000-fold increase (Figure 3a,b). In the fetal-derived HAE that was apically infected, the amount of viral RNA increased in the apical compartment, but this was lower for all three viruses compared to the viral RNA measured in the adult HAE (Figure 3c). No viral RNA could be detected in the basolateral compartment of the fetal HAE after infection (Figure 3d). Upon basolateral infection, no replication was detected via RT-qPCR for all three viral strains (Appendix A). In order to assess whether the shed viral particles were infectious, a TCID50 assay was performed. All three viruses, when infected apically, showed significant production of the infectious viral particles at 24 hpi in the adult HAE while CVA-13 also showed a significant increase in the fetal HAE (Appendix A).

### 3.3. CVA-13, CVA-20, and EV-C99 Infect Ciliated Cells in the Airway Epithelium

We next performed immunostaining of the infected adult HAE cultures at 8 h post-infection, which is the time that picornaviruses require to complete one replication cycle [45]. CVA-13, CVA-20, and EV-C99 staining co-localized with ciliated cells were identified using the β-tubulin antibody (Figure 4a). None of the EV-C viruses analyzed, identified via the viral protein, co-localized with secretory goblet cells showed as mucin 5AC positive cells (Figure 4b), or the basal cells identified via the p63 antibody (Figure 4c). Interestingly, we observed more goblet cells in the infected conditions (CVA-13 and CVA-20) than in the control, even though this cell type was not the target of infection for these viruses (Figure 4b).

### 3.4. CVA-13, CVA-20 and EV-C99 Replicate in the Human Intestinal Epithelial (HIE) Model

In order to study infection in the intestine, the viral infection of the HIE cultures was carried out with CVA-13, CVA-20, and EV-C99. In the apically infected adult-derived HIE, an increase of a 100–1000 fold in viral RNA could be detected in the apical compartment for CVA-13, CVA-20, and EV-C99 (Figure 5a). No increase in viral RNA was observed basolaterally (Figure 5b). In the fetal-derived HIE, we observed the replication of all three viruses upon apical infection (Figure 5c). However, the increase detected in RNA copies was slightly lower than the one measured in the adult-derived intestinal tissue, consistent with findings in the HAE cultures. Viral RNA in the basolateral side was only detected for EV-C99 infection in fetal HIE (Figure 5d). The infection of fetal and adult HIE was also conducted by inoculating the virus in the basolateral compartment. However, this did not result in any replication for any of the viruses tested, whether the sampling was performed apically or basolaterally (Appendix A). A TCID50 assay demonstrated that apical infection with CVA-13, CVA-20, and EV-C99 produced infectious viral particles at 24 hpi, which were still present at 120 hpi (Appendix A).

### 3.5. CVA-13 and CVA-20 Infect Enterocytes While EV-C99 Infects Both Enterocytes and Enteroendocrine Cells

The immunostaining of the infected adult HIE cultures at 8 h post-infection was performed. CVA-13, CVA-20, and EV-C99 VP1 localized with the enterocytes as shown with villin staining (Figure 6a). EV-C99 also infected enteroendocrine cells, marked with chromogranin (ChgA) (Figure 6b). Finally, proliferating cells, stained with a PCNA antibody, were not observed to be targeted by any of the three viruses tested (Figure 6c).

## 4. Discussion

In this study, we showed the replication of non-polio EV-C genotypes CVA-13, CVA-20, and EV-C99, isolated from the stool samples, in both the human airway and intestinal epithelial models with a higher efficiency in adult-derived cultures compared to fetal-derived cultures. The infection route, the shedding route, and the fold change in viral RNA copies for each virus in each model are summarized in Appendix A. The replication of all three viruses only occurred upon apical infection in all the models used. This is in contrast with what is known, from the mouse studies for poliovirus, which passes via M cells and infects the intestinal epithelium basolaterally [22]. This highlights the importance of further characterizing non-polio EV-C in human models. CVA-13- and CVA-20-infected adult HAE and adult HIE more efficiently than the fetal counterparts, and the virus was mainly shed in the apical compartment. EV-C99 also showed a very efficient replication in the adult HAE, but basolateral shedding was detected more than apical shedding. In the intestinal models, EV-C99 replicated as efficiently in the adult as in the fetal-derived epithelium. In both cases, the virus was predominantly shed apically.

Non-polio EV-C genotypes are known to be hard-to-grow viruses [46,47,48]. Therefore, very few studies are available reporting the propagation of these viruses. In particular, one study reported the isolation of CVA-13 and CVA-20, among other Coxsackievirus A, on human amnion cells [47] while another study described the isolation of several Coxsackievirus A on RD99 cells and guinea pig embryo (GPE) cells [48]. However, in both cases, cells were inoculated with isolates obtained from previously infected suckling mice, which showed paralysis as an indication of successful infection, as these animal models were known to effectively support infection with these viruses. Despite this first infection in the susceptible models, several passages on the cell lines were still required to obtain the maximal CPE in the cell cultures, resulting in an increased chance of adaptions as demonstrated by the significant decrease in virulence observed in mice [47]. We showed successful isolation in only two passages on Caco2 cells for the three EV-C types by directly inoculating the clinical material of stool origin on the cell lines. Caco2 cells were highly susceptible to infection with CVA-13, CVA-20, and EV-C99. Moreover, we were also able to obtain replication after culturing the stool samples of CVA-13 and EV-C99 directly on the HAE cultures (data not shown). When comparing the full-length sequence of the virus cultured in the primary cultures to the original fecal material, no mutations were observed for CVA-13 while four were observed for EV-C99, but none in the VP1 region. This suggests that primary cultures may be useful for isolating viruses without introducing too many mutations; however, their utility for downstream applications is limited due to the low volume of material that is obtainable.

The effort to obtain clinical isolates rather than lab-adapted isolates allowed us to study viruses in the closest form possible to the one that has caused infection in human patients. We aimed for this as differences in replication kinetics have been reported depending on the viral stock used [49]. Thus, we used clinical isolates obtained after only 2 passages on Caco2 cells. Despite minimal passaging, we observed differences in passage 2’s isolates compared to the original material. Some of these point mutations, particularly in VP1, could impact viral replication or cell tropism. Critical (point) mutations resulting in more severe pathogenesis [50], in increased binding [51], or in a different receptor usage [52] have been described for different EV genotypes. However, similar mutations have not been reported for non-polio EV-C, making it difficult to conclude whether these few mutations that we observed possibly affected the mechanism of action or cell tropism.

CVA-13, CVA-20, and EV-C99 infected both the human airway and gut epithelium, indicating that both tissues may serve as entry and replication sites for non-polio EV-C using a fecal–oral route of transmission. Although CVA-13, CVA-20, and EV-C99 are mainly found in feces samples showing gastrointestinal symptoms, we showed that the human airway is also susceptible to infection. We observed consistently lower-fold changes in fetal-derived tissues compared to adult-derived tissues. Thus, these differences in viral replication efficacy between fetal and adult tissues did not explain the preferential infection of children via EV-C. Our results could be due to the technical differences between the models, such as the differentiation stage of the cells. Alternatively, it could suggest that viral replication kinetics and tropism are not the major factors contributing to the high prevalence of these viruses observed in children.

Finally, we showed that ciliated cells are the main target of infection for CVA-13, CVA-20, and EV-C99 in the human airway epithelium. They are the most common cell type, accounting for 50 to 80% of the epithelial cells present in the human airway [53]. The same cell tropism has been shown for other respiratory EVs such as EV-D68 and EV-D94 [54]. However, no evident damage or depletion of ciliated cells was observed upon infection with CVA-13, CVA-20, and EV-C99, unlike what was shown for EV-D68. [35,54]. Interestingly, a higher number of goblet cells was observed after infection as compared to the control. It is possible that the increase in goblet cells occurs in response to infection since increased mucus secretion from these cells helps in protecting the epithelium from further infection. However, it would be worthwhile to further investigate this aspect in the future.

In the intestinal epithelium, these viruses target the enterocytes, which are the most abundant cell type in the small intestine. The same cell tropism was previously observed for EV-A71 and CVA-16, which belong to EV species A [55]. EV-C99 also infected the enteroendocrine cells. These secretory cells have been shown to be the target of infection of another enterovirus of the B species, Echovirus 11 (E11), in the intestine [56]. Despite being studied for many years, poliovirus cell tropism remains elusive. There are different hypotheses of poliovirus infecting lymphatic tissue, M-like cells, lamina propria, macrophages, and dendritic cells [22]. Only one study showed the presence of poliovirus in enterocytes but not in the M cells of transgenic mice [57].

In conclusion, we cultured the clinical isolates of CVA-13, CVA-20, and EV-C99 which allowed us, for the first time, to characterize the tissue and cell tropism of these understudied viruses. Using human intestinal and airway models, we showed that CVA-13, CVA-20, and EV-C99 replicated in both tissues despite being mainly associated with gastrointestinal infections in humans. These viruses displayed specific tropism for ciliated cells in the airway while they infected enterocytes in the intestine. EV-C99 infected both enterocytes and enteroendocrine cells in the intestine. With this study, we have taken a first step in defining the tropism of infection of the three most prevalent non-polio EV-C. Further characterization, such as an innate immune response, is needed for an increased understanding of these viruses for which an effective treatment is still missing [58,59] and that can threaten poliovirus eradication due to inter-species recombination with poliovirus. Organotypic cultures have the potential to be used in a co-culture set up with immune cells to address these questions [60,61]. This highlights the versatility of these human-based models, which can further be exploited to study host–pathogens interactions.

## Figures and Tables

**Figure 1 viruses-15-01823-f001:**
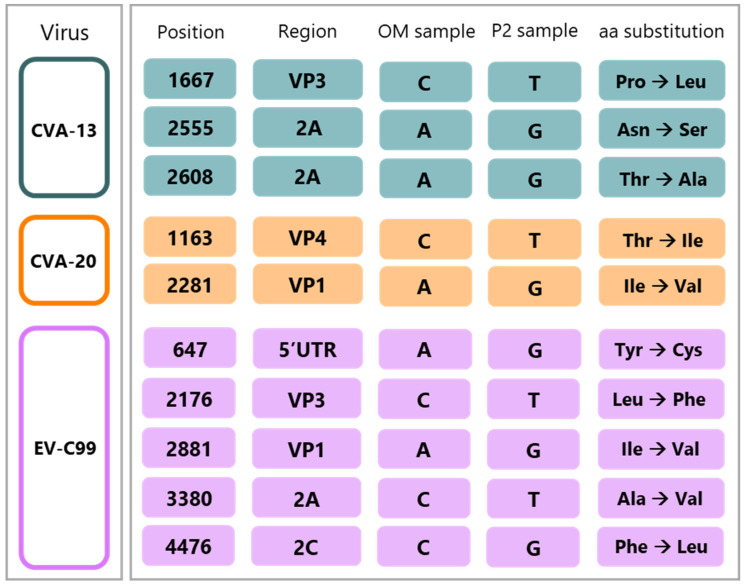
Mutations acquired by CVA-13, CVA-20, and EV-C99 after two passages on Caco2 cells. For each mutation, the nucleotide position, the region of the genome, the nucleotide in the OM sample, the nucleotide in the P2 sample, and the amino acid substitution are indicated. Green indicates CVA-13, orange indicates CVA-20, and pink indicates EV-C99. **CVA-13** (Coxsackievirus 13), **CVA-20** (Coxsackievirus 20), **EV-C99** (Enterovirus C 99), **OM sample** (original fecal sample), **P2** (passage 2), **aa** (amino acid), **5′UTR** (5′ untranslated region), **VP3** (capsid protein 3), **2A** (2A protease), **VP4** (capsid protein 4), **VP1** (capsid protein 1), **2C** (2C protein), **G** (guanine), **A** (adenosine), **C** (cytosine), **T** (thymine).

**Figure 2 viruses-15-01823-f002:**
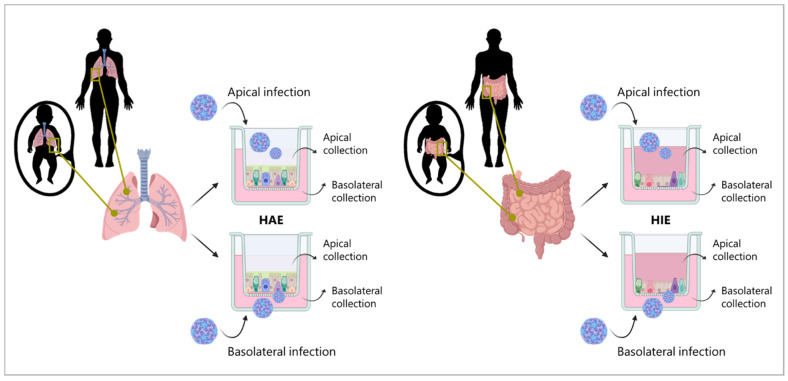
Schematic representation of the experimental layout. Airway and intestinal tissues are isolated from both fetal and adult donors. Monolayers of each tissue and each donor are then generated. Infection is performed both apically and basolaterally and supernatant is collected at specified time points from both compartments. **HAE** (human airway epithelium), **HIE** (human intestinal epithelium).

**Figure 3 viruses-15-01823-f003:**
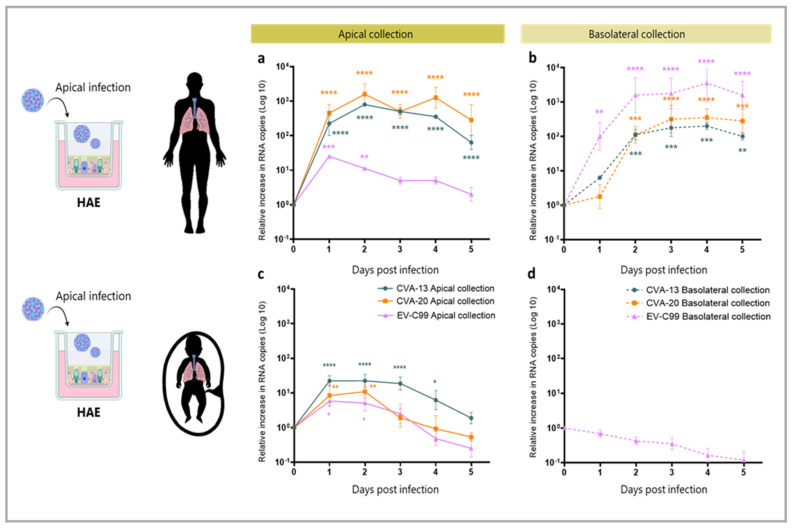
Replication of CVA-13, CVA-20, and EV-C99 on fetal and adult-derived airway monolayers when apically infected. Viral load, plotted as Log 10, of CVA-13, CVA-20, and EV-C99 detected in supernatant samples via RT-qPCR and expressed as relative increase in RNA copies compared to time 0 h. (**a**) Replication kinetics in the apical compartment upon apical infection of adult-derived HAE. (**b**) Replication kinetics in the basolateral compartment upon apical infection of adult-derived HAE. (**c**) Replication kinetics in the apical compartment upon apical infection of fetal-derived HAE. (**d**) Replication kinetics in the basolateral compartment upon apical infection of fetal-derived HAE. The data represent mean ± SEM of two technical replicates of HAE inserts made from a pool of 14 donors (panel **a**) and of three biological replicates (panel **b**–**d**) with three technical replicates for each. The solid line indicates viral RNA in the apical compartment (apical collection) and the dotted line indicates viral RNA in the basolateral compartment (basolateral collection) of CVA-13 (blue), CVA-20 (orange), and EV-C99 (purple). * *p*-value < 0.05, ** *p*-value < 0.01, *** *p*-value < 0.001, **** *p*-value < 0.0001.

**Figure 4 viruses-15-01823-f004:**
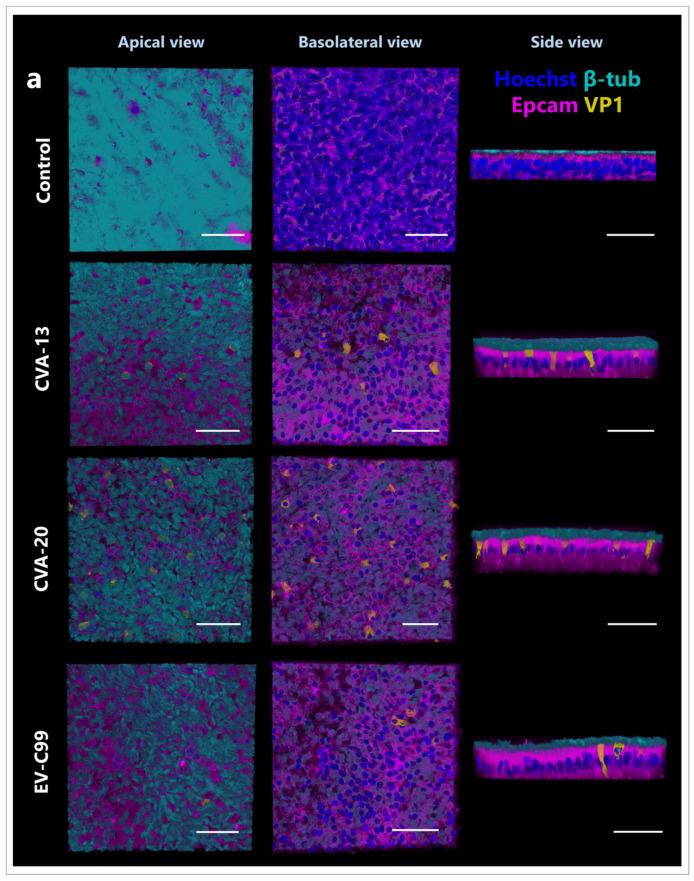
Immunofluorescence images of HAE infected with CVA-13, CVA-20, and EV-C99. Three-dimensional confocal images from an apical, basolateral, and side view at 8 h post-infection is shown for all viruses. (**a**) Ciliated cells are stained with β-tubulin (cyan), nuclei are stained with Hoechst 33342 (blue), cell boundaries are visualized by staining for Epcam (magenta), and viruses are stained for capsid protein VP1 (yellow). (**b**) Goblet cells are stained for Muc5AC (cyan), nuclei are stained with Hoechst 33342 (blue), cell boundaries are visualized with Epcam (magenta), and viruses are stained for capsid protein VP1 (yellow). (**c**) Basal cells are stained for p63 (light blue), nuclei are stained with Hoechst 33342 (blue), tight junctions are stained for zo-1 (magenta), and viruses are stained for capsid protein VP1 (yellow). The scale bar is 50 µm.

**Figure 5 viruses-15-01823-f005:**
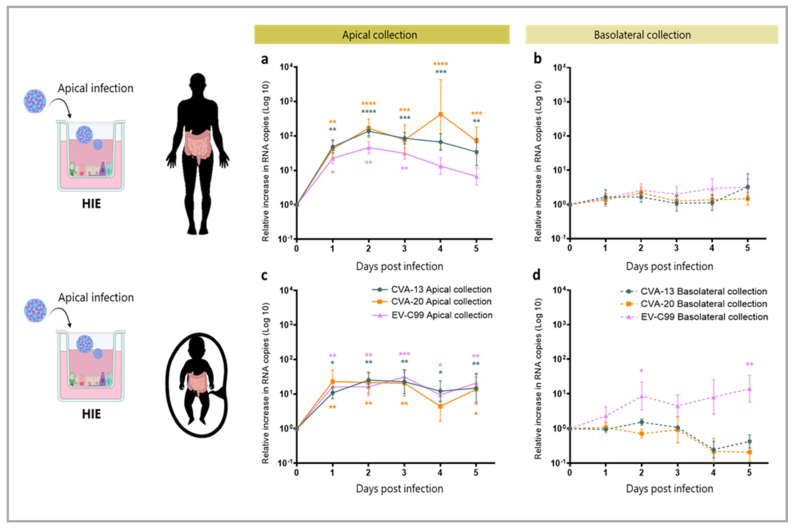
Replication of CVA-13, CVA-20, and EV-C99 on fetal and adult-derived intestinal monolayers when apically infected. Viral load, plotted as Log10, of CVA-13, CVA-20, and EV-C99 detected in supernatant samples via RT-qPCR and expressed as relative increase in RNA copies compared to time 0 h. (**a**) Replication kinetics in the apical compartment upon apical infection of adult-derived intestinal monolayer. (**b**) Replication kinetics in the basolateral compartment upon apical infection of adult-derived intestinal monolayer. (**c**) Replication kinetics in the apical compartment upon apical infection of fetal-derived intestinal monolayer. (**d**) Replication kinetics in the basolateral compartment upon apical infection of fetal-derived intestinal monolayer. The data represent mean ± SEM of three biological replicates with three technical replicates for each. The solid line indicates viral RNA in the apical compartment (apical collection) and the dotted line indicates viral RNA in the basolateral compartment (basolateral collection) of CVA-13 (blue), CVA-20 (orange), and EV-C99 (purple). * *p*-value < 0.05, ** *p*-value < 0.01, *** *p*-value < 0.001, **** *p*-value < 0.0001.

**Figure 6 viruses-15-01823-f006:**
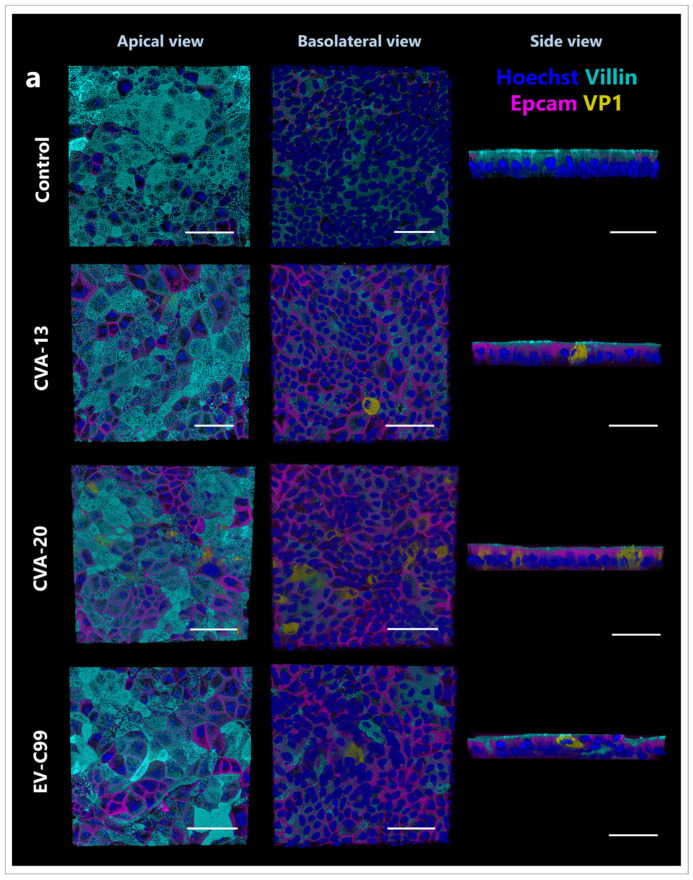
Immunofluorescence images of HIE infected with CVA-13, CVA-20, and EV-C99. Three-dimensional confocal images from an apical, basolateral, and side view at 8 h post-infection for all viruses. (**a**) Enterocytes are stained for villin (cyan), nuclei are stained with Hoechst 33342 (blue), cell boundaries are visualized by staining for Epcam (magenta), and viruses are stained for capsid protein VP1 (yellow). (**b**) Enteroendocrine cells are stained for ChgA (cyan), nuclei are stained with Hoechst 33342 (blue), tight junctions are stained for zo-1 (magenta), and viruses are stained for capsid protein VP1 (yellow). (**c**) Proliferating cells are stained for PCNA (cyan), nuclei are stained with Hoechst 33342 (grey), cell boundaries are visualized using Epcam (magenta), and viruses are stained for capsid protein VP1 (yellow). Arrows indicate some PCNA positive cells. The scale bar is 50 µm.

## Data Availability

The raw data supporting the conclusions of this article will be made available by the authors, upon request, without undue reservation. The data will be available on Figshare.com.

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
