# Peer review of "Non-Polio Enterovirus C Replicate in Both Airway and Intestine Organotypic Cultures"

_viruses, 2023, doi:10.3390/v15091823_

Round 1

Reviewer 1 Report

In this manuscript, Moreni et al. assesses infection of very low-passage clinical isolates of typically hard-to-grow non-polio EV-C viruses in both fetal and adult organoids representing the airway and intestinal epithelium. Their data demonstrate the polarity of virus infection and shedding as well as cell tropism in these models. The experiments are carefully performed, the manuscript is very well-written, and figures very clearly presented. Further, the authors should be commended on the quality of their immunofluorescent staining and microscopy images. Overall, the work they have performed is fundamental in understanding the basic infection parameters and cell tropism of these viruses. 

I have only minor comments on the existing work (points 1, 2, 4, and 7 below), but suggest several ways the authors may take advantage of their system (or even perhaps already-existing samples) to increase the impact of their paper:

1)    Page 9, line 341.  This sentence is slightly confusing as is sounds like “even though these cells were not infected” is referring to the control, not goblet cells in particular.  Please clarify.

2)    For consistency in labeling and to make comparisons easier for the reader, I would suggest to change the labels in Supplemental Figure 2 from hours to days post infection (as done in the other figures).

3)    The increase in goblet cells in Figure 4 was quite striking, especially at this early time point. It would be very interesting to know if this is even more pronounced at later time points post-infection.  Similarly, does the frequency of infected cells change over time?

4)    In Figure 6 the authors comment that none of the three viruses target proliferating cells. However, I do not see any proliferating cells (PCNA+) in the EV-C99-infected culture.  Could the authors use white arrows to highlight these cells here, or select another image that has more obvious PCNA+ cells? 

5)    The authors note there was “no evident damage or depletion of ciliated cells…upon infection”.  Could this be further substantiated by measuring TEER or by performing a cytotoxicity assay using the apical washes taken already for titer determination?  Here again, later time points would also be informative as cytopathic effects may not be obvious at 8 hpi.

6)    Cytokine analysis on basolateral supernatants would also add to the dataset and help to further characterize these viruses and may also provide some insight into the underlying mechanism leading to increased goblet cell numbers.

7)    The sourcing of fetal tissues for this study is not entirely clear in the methods section.  If additional regulations or permissions are required here, they should be noted.

Reviewer 2 Report

In this manuscript, the authors cultured clinical isolates of CVA-13, CVA-20, and EV-C99 to characterize the tissue and cell tropism of these viruses. Clinical isolates are closer form caused infection in human patients possible rather than lab-adapted isolates. They used human airway and intestinal organotypic models to evaluate the difference in the infection route, the shedding route, and the change in viral RNA copies. Notably, they found ciliated airway cells and enterocytes are the target of infection for all three viruses, as well as enteroendocrine cells for EV-C99 by immunostaining assay during differentiated cell types contained in these cultures. Moreover, they compared infection in fetal-derived organotypic models with adult-derived organotypic models in efficacy or tropism of infection.

Some issues require further clarification or modification as shown below:

1.      Page 7, line 295: “To investigate whether the higher prevalence of non-polio EV-C in infants compared to adults was due to a more pronounced susceptibility of neonatal airway, we inoculated both fetal, as a neonatal surrogate, and adult-derived HAE cultures.” Is there sufficient evidence of fetal regard as a neonatal surrogate (e.g. whether the physiological characteristics of the two are consistent)?

2.      As shown in Figure 3 and Supplementary Figure S1, the Y-axis were “Relative increase in RNA copies (Log 10)”, please check, the unit of the Y-axis should be actually the "RNA copies (Log 10)" but not the relative increase.

3.      Please check the whole text and revise the language, such as: Page 18, line 409: It would be better to changeeach virus in each model are summarized” toeach virus in each model is summarized”; Page 19, line 476: It would be better to changeThere are different hypothesis” toThere are different hypotheses”.

English Language is OK, minor revision needed.

Author Response

1) We thank the reviewer for the comment. As far as we are aware, there has not been any extensive comparison of the phenotypes of fetal and neonatal cultures. The lack of surgical neonatal material is a bottleneck for such a study and is a key reason for the use of fetal material in our study. As we are working with model systems, there are some implicit assumptions such as the one above and thus, the data from these models should interpreted within these limitations. We do not imply to extrapolate this data directly to neonatal infections but as of now, the fetal model is closest working model for neonatal infections. 

2) We thank the reviewer for the comment. “Relative increase” refers to the fact that data are normalized to time point 0h, so it is relative to time 0h. While “(Log 10)” refers to the fact that Log 10 values are plotted. So, we believe the way we indicated is correct. We added “plotted as Log 10” in the legend for clarity (page 8, line 325 and page 13, line 375).

3) We thank the reviewer for the comment. We have changed the sentences the way it was suggested (now page 18, line 413 and page 19, line 481).

Reviewer 3 Report

The authors isolated CVA-13, CVA-20, and EV-C99 from stool samples. They showed that the isolates efficiently infect and replicate in human airway and intestinal epithelial organotypic cultures. They further showed that ciliated airway cells and enterocytes are the target of infection for all three viruses, whereas enteroendocrine cells are targets for EV-C99.  The study is clear. The paper is well written. Perhaps adding couple of sentences in the discussion section on potential application of the model systems could i9ncrease impact and attract more attention.

Author Response

We thank the reviewer for the comment. We added a couple of sentences at the end as suggested (page 19, lines 496-498).